

# Genome-wide analysis of the heat shock transcription factor family reveals saline-alkali stress responses in *Xanthoceras sorbifolium*

Lulu Li[1], Yiqian Ju[1], Cuiping Zhang[1], Boqiang Tong[2], Yizeng Lu[2], Xiaoman Xie[2] and Wei Li[1]

[1] Qingdao Agricultural University, Qingdao, China
[2] Shandong Provincial Center of Forest and Grass Germplasm Resources, Jinan, China

Corresponding authors
Xiaoman Xie, xxm529@126.com
Wei Li, weili@qau.edu.cn

## ABSTRACT

The heat shock transcription factor (HSF) family is involved in regulating growth, development, and abiotic stress. The characteristics and biological functions of *HSF* family member in *X. sorbifolium*, an important oil and ornamental plant, have never been reported. In this study, 21 *XsHSF* genes were identified from the genome of *X. sorbifolium* and named *XsHSF1-XsHSF21* based on their chromosomal positions. Those genes were divided into three groups, A, B, and C, containing 12, one, and eight genes, respectively. Among them, 20 *XsHSF* genes are located on 11 chromosomes. Protein structure analysis suggested that XsHSF proteins were conserved, displaying typical DNA binding domains (DBD) and oligomerization domains (OD). Moreover, HSF proteins within the same group contain specific motifs, such as motif 5 in the HSFC group. All *XsHSF* genes have one intron in the CDS region, except *XsHSF1* which has two introns. Promoter analysis revealed that in addition to defense and stress responsiveness elements, some promoters also contained a MYB binding site and elements involved in multiple hormones responsiveness and anaerobic induction. Duplication analysis revealed that *XsHSF1* and *XsHSF4* genes were segmentally duplicated while *XsHSF2, XsHSF9,* and *XsHSF13* genes might have arisen from transposition. Expression pattern analysis of leaves and roots following salt-alkali treatment using qRT-PCR indicated that five *XsHSF* genes were upregulated and one *XsHSF* gene was downregulated in leaves upon NaCl treatment suggesting these genes may play important roles in salt response. Additionally, the expression levels of most *XsHSFs* were decreased in leaves and roots following alkali-induced stress, indicating that those XsHSFs may function as negative regulators in alkali tolerance. MicroRNA target site prediction indicated that 16 of the *XsHSF* genes may be regulated by multiple microRNAs, for example *XsHSF2* might be regulated by miR156, miR394, miR395, miR408, miR7129, and miR854. And miR164 may effect the mRNA levels of XsHSF3 and XsHSF17, XsHSF9 gene may be regulated by miR172. The expression trends of miR172 and miR164 in leaves and roots on salt treatments were opposite to the expression trend of *XsHSF9* and *XsHSF3* genes, respectively. Promoter analysis showed that *XsHSFs* might be involved in light and hormone responses, plant development, as well as abiotic stress responses. Our results thus provide an overview of the HSF family in *X. sorbifolium* and lay a foundation for future functional studies to reveal its roles in saline-alkali response.

## INTRODUCTION

Heat shock transcription factors (HSFs) are widely expressed, including in bacteria, fungi, yeast, animals, and plants (*Wiederrecht, Seto & Parker, 1988*; *Clos et al., 1990*; *Rabindran et al., 1991*; *Sarge et al., 1991*; *Scharf et al., 1991*; *Ghorbani, Alemzadeh & Razi, 2019*). However, the number of *HSF* genes varies greatly in various classes of organisms, including between animals and sessile plants. Drosophila and vertebrates have only one and three HSF genes, respectively, while *Arabidopsis*, rice, tomato, and soybean contain 21, 18, 23, and 38 HSF genes, respectively (*Guo et al., 2008*; *Hübel & Schöffl, 1994*; *Yang et al., 2016*; *Li et al., 2014*). The HSFs possess the general structure of transcription factors: a DNA binding domain (DBD), an oligomerization domain (OD), and a nuclear localization signal (NLS). Furthermore, some HSFs are endowed with a nuclear export signal (NES) and a C-terminal activation domain (CTAD) (*Harrison, Bohm & Nelson, 1994*; *Schultheiss et al., 1996*; *Kotak et al., 2004*). Among these structures, the DBD and OD are the most conserved domains. The DBD is located at the N-terminus and has a helix-turn-helix hydrophobic structure, which is required to recognize and bind the heat shock element (HSE) motif in the promoters of target genes (*Schultheiss et al., 1996*). The OD domain is composed of two hydrophobic heptapeptide repeat regions A and B (HR-A and HR-B), forming a coiled-coil structure that helps HSF proteins form homotrimers (*Peteranderl et al., 1999*). Based on the number of amino acid residues inserted between HR-A and HR-B, HSFs can be divided into three types: HSFA with 21 amino acid residues inserted, HSFC with seven amino acid residues inserted, and HSFB with no amino acid residue inserted in this region (*Nover et al., 1996*). The NLS and NES domains guide the nuclear import and export, respectively, of HSF proteins (*Lyck et al., 1997*; *Heerklotz et al., 2001*; *Kotak et al., 2004*). The combined effect of the NLS and NES determines the subcellular localization of HSFs in different states.

The first report on plant HSFs dealt with tomatoes under heat stress, but subsequent studies have shown that HSFs are involved in a variety of stress responses including responses to freezing, drought, and salinity-alkalinity tolerance (*Nover et al., 2001*; *Garg et al., 2002*; *Scharf et al., 1991*). There are many studies on the role of *HSF* genes to heat stress in plant. Notably, the members of the HSF A class play important roles in heat stress response in *Arabidopsis* (*Busch, Wunderlich & Schöffl, 2005*), tomato (*Li et al., 2013*), wheat (*Harsh et al., 2013*) and strawberry (*Hu et al., 2015*). HSFs also play important roles in regulating the resistance of plants to low temperatures, such as *AtHSFA6a*, *AtHSFA6b*, *AtHSFA9*, *AtHSFC1*, *OsHSFC1*, *OsHSFC2b*, *OsHSFA3*, *OsHSFA4d*, *OsHSFA7*, *OsHSFA9*, *BraHSF043*, and *BraHSF039* from *Arabidopsis*, rice, and cabbage (*Miller & Mittler, 2006*; *Liu et al., 2010*; *Huang et al., 2015*), (*Scharf et al., 1991*; *Ma et al., 2014*). Moreover, several HSF genes are involved in the adaptive response of plants to drought (*Wiederrecht, Seto & Parker, 1988*; *Yoshida, Nishizawa & Yabuta, 2006*; *Guo, Wang & Zhang, 2017*; *Ghorbani,*

*Alemzadeh & Razi, 2019*). Overexpression of either *AtHSFA6a* or *AtHSFA1b* enhances tolerance for drought stress in *Arabidopsis* (*Bechtold et al., 2013*). but some HSF genes were identified that negatively impact drought resistance in plants, such as *OsHSFB2b* in rice (*Xiang et al., 2013*).

Similar to drought stress, the expression patterns of HSF family members varied during different stages of salt treatment. In the strawberry leaves treated with 300 mmol $L^{-1}$ NaCl, the expression of *FvHsfA2a* increased during the early stage of stress, while the expression of *FvHsfA3a*, *FvHsfA5a,* and *FvHsfA9a* increased during the middle or late stages (*Hu et al., 2015*). Likewise, different members of the HSF family can act as positive or negative regulators in salt stress response. Notably, expression of *AtHsfA2*, *OsHsfA7*, *OsHsfA2e*, and *TaHsfA2d* enhances salt tolerance in *Arabidopsis*, rice, and wheat, respectively (*Ogawa, Yamaguchi & Nishiuchi, 2007*; *Yokotani et al., 2008*; *Liu et al., 2013*). However, over expression of the *OsHsfB2b* gene reduced the salt tolerance of rice (*Xiang et al., 2013*).

Due to the important roles of *HSFs* in responding to both biological and abiotic stresses in plants, numerous studies have concentrated on their regulatory mechanism. The HSPs, whose transcription is regulated by *HSFs,* accumulate rapidly in plants to help related proteins fold, intracellular distribution and degrade under environmental stress, when stress causes the occurrence of protein damage or inactivation (*Fragkostefanakis et al., 2014*). Although *HSFs* can participate in a variety of stresses, its transcriptional regulation mechanism in response to heat stress is relatively well studied. HSFs can regulate plant responses to stress by interacting with HSP proteins, other HSFs, or other substances. In a normal environment, the AtHSFA2 protein is distributed in the cytoplasm and nucleus with a low activity. In *Arabidopsis* AtROF1 and AtHSP90.1 form a complex that is stored in the cytoplasm. Under heat stress, the ROF-HSP90.1 complex interacts with HSFA2, facilitating its translocation to the nucleus, thereby regulating the expression of heat-responsive genes (*Yokotani et al., 2008*). Similarly, SlHSFA1 can interact with SlHSFA2 to activate the transcriptional expression of *SlHSP90* and *SlHSP70* in potato (*Solanum lycopersicum*) (*Scharf et al., 1998*; *Kwan et al., 2009*). Regardless of the fact that SlHSFB1 protein is not active, it can help SlHSFA1 to play a regulatory role in the heat shock process (*Bharti et al., 2004*). In addition, SlHSP90 can, in turn, regulate the expression of SlHSFA2 and SlHSFB1 proteins, thus forming a negative feedback loop (*Hahn et al., 2011*). HSF transcription factors form complexes to participate in stress signal perception and transmission processes. Light and other stresses can induce the accumulation of reactive oxygen species (ROS) in *Arabidopsis*. HSFA4, as a receptor of the $H_2O_2$ signal, can transmit the ROS signal to downstream transcription factors, including ZATs and WRKYs, through the mitogen-activated protein kinase (MAPK) pathway, thus triggering the cellular response network (*Davletova et al., 2005*; *Qu et al., 2013*). Furthermore, the transcriptional activity of several *HSFs* is regulated by calcium signals, triggered by heat in plasma membrane of the cell (*Liu et al., 2008*; *Suzuki et al., 2011*). The HSFs are functionally diverse and display different expression levels in different plant tissues and under different stress conditions. However, little is known about the regulatory mechanisms of HSFs in numerous species except model plants such as tomato and *Arabidopsis*. Furthermore, most of the studies on

HSFs focus on heat stress, and their roles in various abiotic stresses, such as salt-alkali, freezing, and humidity, are still lacking.

*Xanthoceras sorbifolium* is an endangered and single species of the genus *Xanthopanax* in the *Sapindaceae.* The wild population is distributed in a ravine of the Loess Plateau in China, having developed root system with resistance to barrenness, drought, and saline-alkali environments (*Bi & Guan, 2014*). Due to its oil-rich seeds as well as its dense and beautiful flowers, the domesticated population had been grown in northern China (*Yu et al., 2017*). However, an excessive salt concentration in the soil remains a limiting environment factor for the cultivation of *X. sorbifolium*. How much saline-alkali stress it can withstand and the mechanism of salt-alkali resistance in *X. sorbifolium* remains unclear. The HSF transcriptional factors not only control plant growth and development but also enhance the resistance to abiotic stress. However, HSFs in *X. sorbifolium* have not been analyzed in detail. A high-quality sequence of the *X. sorbifolium* genome has been published, which provides the foundation for genome-wide analysis of the *HSF* genes (*Bi et al., 2019*). In this study, 21 *HSF* genes from *X. sorbifolium* were identified and their gene and protein features were characterized in detail. To reveal the mechanism of the role the HSF transcription factor plays in salt-alkali tolerance regulation in *X. sorbifolium*, the expression profiles of 15 *XsHSFs* were examined in the roots and leaves under saline-alkali stress. Therefore, this study provides the functional characteristics of the HSF family in *X. sorbifolium* and lays the foundation for their application in molecular breeding.

## MATERIALS & METHODS

### HSF genes identification

The genomes of *Arabidopsis thaliana* and rice (*Oryza sativa*) were downloaded from Phytozome (https://phytozome.jgi.doe.gov/pz/portal.html) (*Nover et al., 2001*; *Lamesch et al., 2012*). The genomic data of *X. sorbifolium* (*Bi et al., 2019*), *Dimocarpus longan* (*Lin et al., 2017*), and https://www.ncbi.nlm.nih.gov/genome/83651 (*Yang et al., 2019*) were collected from the Giga database (http://gigadb.org/). The candidate genes of the HSF family were extracted from the *X. sorbifolium, D. longan,* and https://www.ncbi.nlm.nih.gov/genome/83651 genome databases using the software HMMER (*Johnson, Eddy & Portugaly, 2010*) based on a Hidden Markov Model (HMM) of the DBD-domain profile (PF00447), obtained from the Pfam database (https://pfam-legacy.xfam.org/). The e-value cut-off was 0.01. To confirm the presence of the DBD domain in candidate XsHSF family members, all of the obtained protein sequences were analyzed by the online CD-search software (http://www.ncbi.nlm.nih.gov/Structure/cdd/wrpsb.cgi). The putative *XsHSF genes* were named depending on their genomic location. The subcellular location of HSF proteins was predicted by the Cell PLoc2 software (*Chou & Shen, 2010*).

### Chromosomal location and gene duplication analyses

The chromosomal location information of *HSF* genes was obtained from *X. sorbifolium* genome database, and the chromosomal location map was produced by TBtools (*Chen et al., 2018*). Tandem duplicated genes were identified and paralogs were separated by five or fewer genes in a 100 kb region, as proposed by *Tu et al. (2010)*. Identification

of segmentally duplicated genes was conducted based on the plant genome duplication database (*Lee et al., 2017*; *Lee et al., 2013*). The non-synonymous substitution (Ka) and synonymous substitution (Ks) values of duplicated pairs were calculated on PAL2NAL (http://www.bork.embl.de/pal2nal) (*Suyama, Torrents & Bork, 2006*) after alignment of protein sequences. The duplication time of duplicated genes was calculated by $T = Ks/2\lambda \times 10^{-6}$ Mya ($\lambda = 1.5 \times 10\text{-}8$ for dicots) (*Blanc & Wolfe, 2004*).

## Gene characteristics, structure, and cis-elements in promoter

Physical parameters of putative XsHSF proteins, including the protein length, molecular weight (kDa), theoretical isoelectric point (pI), and grand average of hydropathicity (GRAVY), were calculated by the ExPASy tool (http://www.expasy.org/tools/protparam.html). The exon-intron structures in each *XsHSF* genes were displayed by alignment of the CDS with DNA sequences using Gene structure display server (GSDS) program (http://gsds.cbi.pku.edu.cn/). The orthologous gene of *XsHSF* in *Arabidopsis* were annotated using the Blast2GO (*Conesa et al., 2005*). The online database PLACE (https://www.dna.affrc.go.jp/PLACE/?action=newplace) was employed to investigate putative cis-regulatory elements in the sequences about 1,500 bp upstream of the transcription start site (ATG).

## Protein domains and conserved motif analyses

The online MEME program was used to display the motif structure of HSF proteins (http://meme-suite.org/tools/meme) on the default parameters except for the optimum motif widths with 6 to 200 residues. SMART (http://smart.embl-heidelberg.de) and CD-search (http://www.ncbi.nlm.nih.gov/Structure/cdd/wrpsb.cgi) web servers were used for structural motif annotation.

## Sequence alignment and phylogenetic analysis

The Clustal X program was used to conduct the multiple sequence alignment of the full-length HSF protein sequences from *X. sorbifolium, D. longan, A. yangbiense, Populus trichocarpa, A. thaliana,* and *O. sativa* (*Thompson et al., 1997*). The Minimum Evolution (ME) phylogenetic tree was constructed by MEGA 6.0 software under the Jones-Taylor-Thornton (JTT) amino acid and Gamma Distributed (G) substitution model. Bootstrap values from 500 replicates were indicated at each node. Finally, the phylogenetic tree was beautified using online software iTOL https://itol.embl.de/itol.cgi.

## MicroRNA target site prediction

The PsRNATarget tool (https://www.zhaolab.org/psRNATarget/) was used to predict microRNA target sites in each full-length *XsHSFs* sequence according to published microRNA sequences from *Citrus sinensis*, belonging to the *Sapindales* (*Song et al., 2012*).

## Preparation of plant materials

*X. sorbifolium* was cultivated in the greenhouse in Qingdao Argriculture University. To explore the expression of *XsHSFs*, seedlings that had been germinating for 30 days were treated with water, 200 mM NaHCO$_3$, or 200 mM NaCl. After 0.5, 6 h, and 7 days of treatments, the roots and leaves of seedlings with different treatments were sampled and

immediately frozen in liquid nitrogen for total RNA extraction and real-time RT-PCR assays. All samples were obtained in three biological replicates. All samples were collected and frozen in liquid nitrogen immediately, then stored in −80 °C for RNA extraction.

## RNA extraction and qRT-PCR

Total RNA was extracted from mature leaves and young roots, separately, on different treatment using the RNA extraction Kit (Tiangen, Beijing, China). A total of 1 μg of total RNA was reverse-transcribed to first-strand cDNA by the PrimerScript RT Reagent Kit (TaKaRa Biotechnology, Dalian, China). Subsequently, qRT-PCR was performed using fluorescent dye SYBR Green I (TaKaRa, Dalian, China) according to the instructions. *GADPH* gene was selected as reference genes. Three biological replicates were carried out and expression levels were calculated by the $2^{-\Delta\Delta Ct}$ method (*Schmittgen & Livak, 2008*). In addition, 500 ng total RNA was used to synthesis miRNA first-strand cDNA by the miRNA 1st strand cDNA synthesis kit (by stem-loop) (Vazyme, China, Nanjing). The 5.8s gene was used as reference gene for miRNA qRT-PCR. Gene primer sequences were listed in Table S1.

# RESULTS

## Identification and characteristic analysis

A total of 21 gene sequences that are part of the HSF family were identified in the *X. sorbifolium* genome using the PF00447 Hidden Markov Model (HMM) sequence as a query (E < 1e$^{-5}$). These sequences were named *XsHSF1-XsHSF21* based on their chromosomal positions from top to bottom successively (Table 1). The number of HSF genes in *X. sorbifolium* was similar to *Arabidopsis* (21), *D. longan* (20), rice (25), *Poplar* (27), whereas it was 0.50-fold less than that in Soybean (52) and 1.4-fold greater than that in *Acer yangbiense* (15) (*Nover et al., 2001*; *Baniwal et al., 2004*; *Wang, Zhang & Shou, 2009*; *Li et al., 2014*). Gene lengths were between 796 to 6,687 bp. The corresponding protein sequences were submitted to CD-search (http://www.ncbi.nlm.nih.gov/Structure/cdd/wrpsb.cgi) web servers to confirm the presence of the DBD domain. The XsHSF proteins ranged in length from 211 and 525 amino acids (aa) with an average of 371.4 aa and the isoelectric points (pI) ranged from 4.75 to 9.03. According to the prediction of the protein subcellular localization by Cell-PLoc 2.0 package, all HSF proteins were located in nucleus (Table 1).

## Chromosomal location and gene duplication

The 20 *XsHSF* genes were distributed on 11 chromosomes (Chr) and no HSF genes were found on Chr2, Chr9, Chr14, and Chr15 (Fig. 1). The *XsHSF21* gene was the only 1 found on an unmapped scaffold. Gene duplication events, leading to gene functional diversity and plant adaptations, are mainly divided into three categories: segmental duplication, tandem duplication, and transposition (*Kong et al., 2007*). We identified 1 pair (*XsHSF1* and *XsHSF4*) in the segmental duplication category and 3 genes (*XsHSF2, XsHSF9,* and *XsHSF13*) that might have arisen from transposition in the *X. sorbifolium* genome, accounting for 23.81% of the genes in the family (Data S1). The Ka/Ks ratios

Li et al. (2023), *PeerJ*, DOI 10.7717/peerj.15929

**Table 1   Characteristics of 21 HSF transcription factors in *X. sorbifolium*.**

| Gene name | Gene ID | Locus | Gene_start | Gene_stop | pI | MW | Strand | Subcellular localization | Gene length (bp) | Protein length (aa) | *Arabidopsis* ortholog |
|---|---|---|---|---|---|---|---|---|---|---|---|
| *XsHSF1* | EVM0009602 | LG1 | 10952517 | 10954672 | 5.6 | 42673.27 | − | Nucleus | 2,156 | 376 | *HSFA7A* |
| *XsHSF2* | EVM0018288 | LG1 | 30144743 | 30149941 | 4.93 | 56493.58 | + | Nucleus | 5,199 | 516 | *HSFA1D* |
| *XsHSF3* | EVM0023941 | LG3 | 5587982 | 5591971 | 5.59 | 56276.6 | + | Nucleus | 3,990 | 502 | *HSFA5* |
| *XsHSF4* | EVM0020189 | LG3 | 25844286 | 25846001 | 5.08 | 43276.34 | + | Nucleus | 1,716 | 375 | *HSF1* |
| *XsHSF5* | EVM0003212 | LG4 | 28823754 | 28824973 | 8.17 | 40264.4 | + | Nucleus | 1,220 | 360 | *HSFB2B* |
| *XsHSF6* | EVM0024360 | LG4 | 28899606 | 28900824 | 8.17 | 40264.4 | + | Nucleus | 1,219 | 360 | *HSFB2B* |
| *XsHSF7* | EVM0006645 | LG5 | 3463810 | 3466132 | 5.34 | 45586.96 | + | Nucleus | 2,323 | 399 | *HSFA4A* |
| *XsHSF8* | EVM0016425 | LG5 | 6794741 | 6795536 | 9.03 | 24596.02 | + | Nucleus | 796 | 211 | *HSF4* |
| *XsHSF9* | EVM0007574 | LG6 | 8861249 | 8867935 | 5.07 | 55755.26 | − | Nucleus | 6,687 | 502 | *HSF3* |
| *XsHSF10* | EVM0007442 | LG6 | 11510725 | 11512056 | 5.55 | 38923.54 | − | Nucleus | 1,332 | 336 | *HSFA1D* |
| *XsHSF11* | EVM0007788 | LG6 | 26645262 | 26646978 | 5.12 | 51340.27 | − | Nucleus | 1,717 | 459 | *HSFA2* |
| *XsHSF12* | EVM0014424 | LG6 | 29759584 | 29760880 | 5.78 | 35230.91 | − | Nucleus | 1,297 | 314 | *HSFC1* |
| *XsHSF13* | EVM0005619 | LG6 | 31542679 | 31546019 | 4.95 | 43166.13 | + | Nucleus | 3,341 | 380 | *HSFA2* |
| *XsHSF14* | EVM0011404 | LG7 | 16687883 | 16688947 | 7.18 | 30429.3 | − | Nucleus | 1,065 | 260 | *HSFB4* |
| *XsHSF15* | EVM0016821 | LG8 | 25452697 | 25456928 | 4.75 | 44388.03 | − | Nucleus | 4,232 | 387 | *HSFA8* |
| *XsHSF16* | EVM0010398 | G10 | 23276109 | 23279230 | 7.01 | 30593.96 | − | Nucleus | 3,122 | 273 | *HSFB2B* |
| *XsHSF17* | EVM0009694 | LG11 | 25864478 | 25866125 | 4.86 | 34933.52 | − | Nucleus | 1,648 | 315 | *HSFB4* |
| *XsHSF18* | EVM0007349 | LG12 | 9980293 | 9981938 | 5.14 | 25548.49 | − | Nucleus | 1,646 | 220 | *HSFB2A* |
| *XsHSF19* | EVM0014086 | LG12 | 24748556 | 24750234 | 5.87 | 36903.25 | − | Nucleus | 1,679 | 330 | *HSFB2B* |
| *XsHSF20* | EVM0010871 | LG13 | 27658322 | 27660823 | 5.15 | 57777.74 | − | Nucleus | 2,502 | 525 | *HSFA3* |
| *XsHSF21* | EVM0015064 | ctg1165 | 8512 | 10872 | 5.34 | 45613.98 | − | Nucleus | 2,361 | 399 | *HSFA4A* |
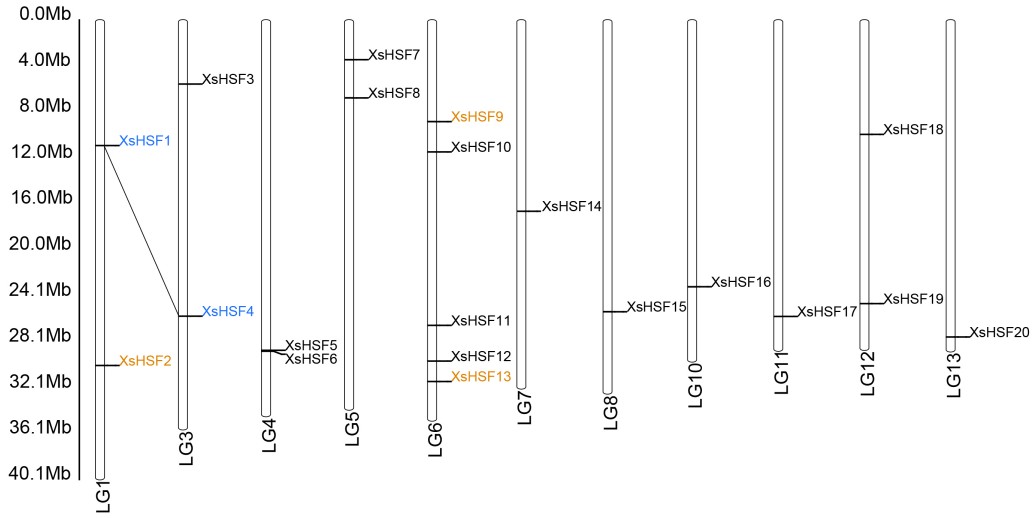

**Figure 1** **Chromosomal location and duplicated gene pairs of *XsHSF* genes.** Blue letters connected with dotted black lines indicate segmental gene pairs, while the orange letters show genes arising from transposition.

of gene pairs were calculated to investigate the gene divergence trend. A Ka/Ks ratio <1 indicates purifying selection. None of the *XsHSF* were randomly distributed.

## Phylogenetic analysis and conservative domain alignment

To reveal the classification of HSF proteins from *X. sorbifolium* and their evolutionary relationships, full-length HSF proteins in *X. sorbifolium, D. longan, Acer yangbiense, Populus trichocarpa, O. sativa,* and *A. thalian* were selected to construct a phylogenetic tree (Fig. 2). The full-length sequences of all HSF proteins identified in this study are presented in Data S2. The HSF proteins of *X. sorbifolium* are allocated into five major groups (Group $\alpha$, $\beta$, $\gamma$, $\delta$, and $\varepsilon$). Group $\alpha$ included eight genes (*XsHSF5, XsHSF6, XsHSF14, XsHSF8, XsHSF16, XsHSF17, XsHSF18, and XsHSF19*), group $\beta$ included four genes (*XsHSF3, XsHSF7, XsHSF10, and XsHSF21*), group $\gamma$ and $\delta$ both have four members, and group $\varepsilon$ only included *XsHSF12*.

## Gene structure and promoter Cis-acting element analysis of *XsHSFs*

The intron/exon arrangements (Fig. 3, left) illustrated the structural similarities of the various *XsHSF* genes. Most of the *XsHSF* genes have one intron in their CDS region except for *XsHSF1* with two introns. The cis-acting elements of the promoter play a leading role in regulating the expression of genes involved in phytohormone and environmental response. The 2,000 bp upstream sequences from ATG of *XsHSF* genes were extracted from *X. sorbifolium* genome and predicted by Plant CARE to identify cis-acting elements. A total of 21 cis-acting elements were identified in the different *XsHSF* promoters (Fig. 3 right, Table S2). Among these predicted cis-acting elements, most were related to light responsiveness, including ACE, G-Box, GT1, 3-AF1 binding site, Sp1, MRE, Box 4, LAMP-element, TCCC-motif, and AE-box. Moreover, a large number of cis-elements related to

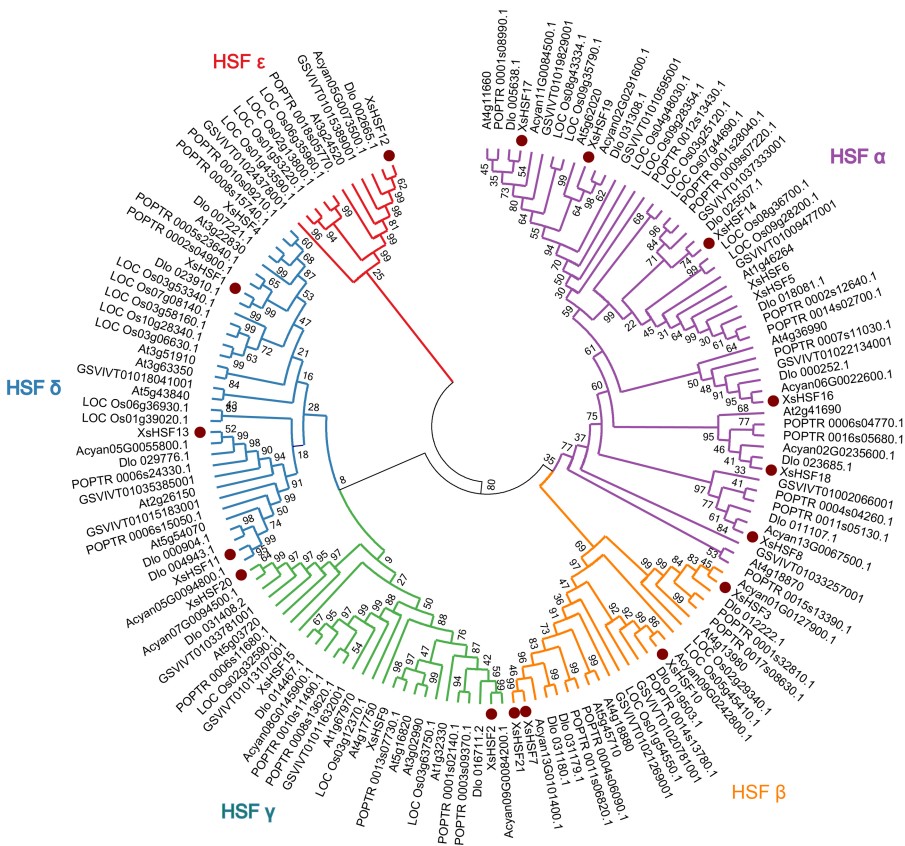

**Figure 2 Phylogenetic tree of HSF transcription factors from Xanthoceras *sorbifolium, Dimocarpus longan, Acer yangbiense, Populus trichocarpa,* Oryza *sativa,* and *Arabidopsis thaliana*.** HSF protein sequences were aligned using Clustal X and the phylogenetic tree was produced using MEGA 6.0 software with the Minimum Evolution method. Bootstrap values are based on 500 replicates. The phylogenetic tree was divided into five groups ( $\alpha$, $\beta$, $\gamma$, $\delta$, and $\varepsilon$ ). Each group is distinguished by a different color.

phytohormones were present on *XsHSF* promoters. For instance, the TGA-element and AuxRR-motif related to auxin responsiveness, the P-box, GARE-motif, and TATC-box related to gibberellin responsiveness, as well as ABRE, which were present in almost all *XsHSF* promoters except those of *XsHSF13*, *XsHSF15*, and *XsHSF20*, were activated in response to abscisic acid, implying that *XsHSF* genes could be induced by phytohormone signals. Various elements, including LTR, MBS, ARE, and TC-rich repeats, that are activated in response to stress responsiveness, were found in *XsHSF* promoters, suggesting that *XsHSFs* may participate in the response to environmental stimulation. Among these, ARE was found in all *XsHSF* promoters except for *XsHSF2*, *XsHSF13*, and *XsHSF18*. In addition, the RY-element, involved in seed-specific regulation, O2-sit, involved in the regulation of zein metabolism, CAT-box, involved in meristem development, and the GCN4-motif, involved in endosperm development, were found in several *XsHSF* promoters, indicating their essential function in plant development.

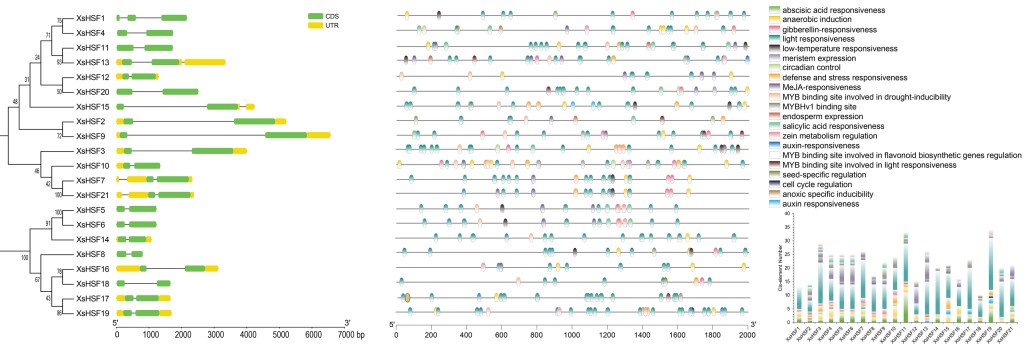

**Figure 3** **Gene structure and promoter cis-element analysis of *XsHSF* in *X. sorbifolium*.** Left, Exon and intron structure of HSFs in *X. sorbifolium*. The orange and green round-corner rectangles represent untranslated region (UTR) regions and exons, respectively, and the black line shows introns. Right, Cis-acting elements of the *XsHSF* promoters.

## Conserved domains and motif analysis

The alignment of the two conservative domains (DBD and OD) is shown in Fig. 4. DBD and OD domains were present in all XsHSF proteins. And XsHSF transcription factors were divided into three classes according to the number of amino acids between HR-A and HR-B in OD domain, XsHSF A subfamily includes members in group $\beta$, $\gamma$, and $\delta$, and XsHSF B and C include the members in group $\alpha$ and $\varepsilon$, respectively. The conserved motifs, analyzed by MEME online tool, are shown in Fig. 5. Each putative motif was annotated using Pfam and SMART softwares. The motifs 1 and 2 in the DBD domain as well as motif 3 in the OD domain were relatively conserved in all XsHSF proteins. However, motif 4 encode NES and are extant only in A class. Motif 6 was present only in the HSF A and B classes but not in the C class. Motif 8 and motif 9 were present in the A and C classes while motif 5 was seen in B class.

## Expression pattern analysis of *XsHSF* genes during salt and alkaline stress

To investigate the roles of *XsHSFs* in response to salt and alkali stress, qRT-PCR analysis was conducted in leaves (Fig. 6) and roots (Fig. 7) and raw data were showed in Data S3. Diverse expression profiles indicated that under normal growth conditions all *XsHSFs* were detected in leaves and roots but that their expression patterns significantly varied. In leaves, the expression of *XsHSF11* and *XsHSF15* was induced by NaCl treatment, while the *XsHSF1*, *XsHSF9*, and *XsHSF7* expression levels were significantly decreased (Figs. 6A, 6E, 6G, 6I, 6K). The trend of *XsHSF3*, *XsHSF8* and *XsHSF14* genes expression was first up and then down, while the expression of *XsHSF5* showed the opposite trend on NaCl stress. Upon NaHCO₃ treatment, except of *XsHSF9* and *XsHSF10*, the expression levels of most genes, including *XsHSF1*, *XsHSF2*, *XsHSF3*, *XsHSF5*, *XsHSF7*, *XsHSF11*, *XsHSF14*, and *XsHSF15*, were significantly decreased in leaves (Fig. 6). In roots, the expression trend of several genes (*XsHSF1*, *XsHSF2*, *XsHSF3*, *XsHSF10*, *XsHSF11* and *XsHSF14* gene) under NaCl treatment was the same as that under NaHCO₃ treatment, *XsHSF1* and *XsHSF10*

![PeerJ]

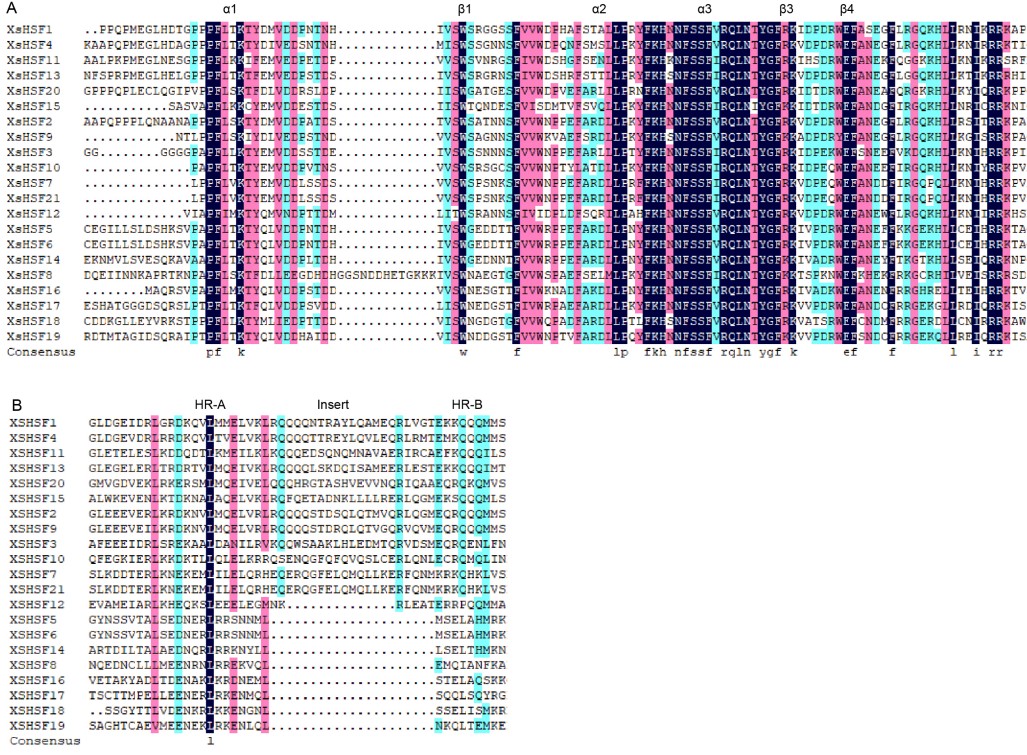

**Figure 4 Multiple sequence alignment of HSF proteins in *X. sorbifolium*.** (A) Multiple sequence alignment and protein sequence logo of the typical DNA binding domain (DBD). (B) Multiple sequence alignment of the oligomerization domain (OD).

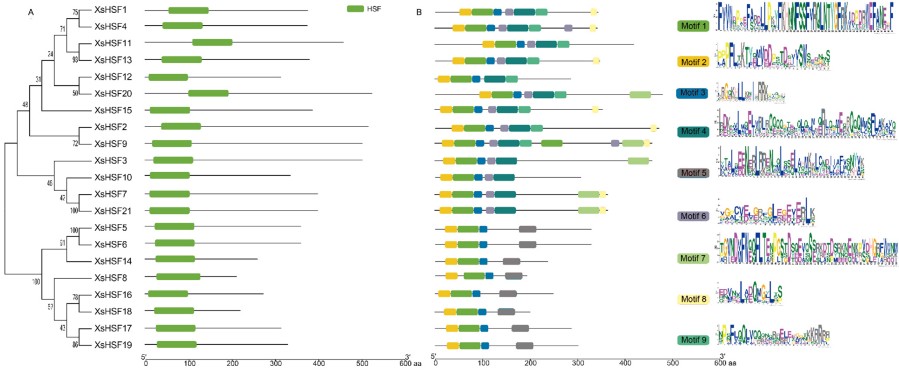

**Figure 5 Conserved domains and motif analysis of XsHSF proteins in *X. sorbifolium*.** Left, Conserved domains analysis; Middle, Conserved motif analysis. The sequence logo (*right*) was created in MEME online software.

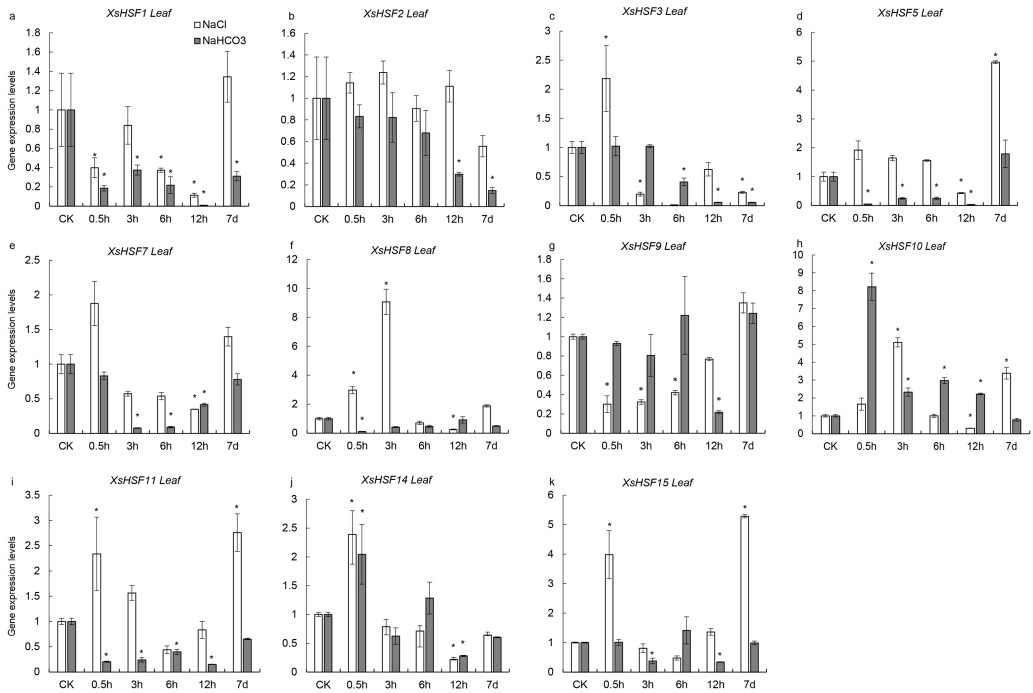

**Figure 6 Expression analysis of ten selected *XsHSFs* in leaves upon different treatments.** *XsGADPH* was used as an internal control. Vertical lines represent S.E. ($n = 3$), ⋆ letters indicate significant differences ($P < 0.05$ by LSD test and fold change $\geq 2$) among treatments. CK, leaf of seedling; $Y$-axis indicate leaf of seedling with 200 mM NaCl or 200 mM NaHCO$_3$ treatment 0.5 h, 3 h, 6 h, 12 h, and 7d, respectively.

were up-regulated, *XsHSF11* and *XsHSF14* were down-regulated (Figs. 7A, 7B, 7C, 7H, 7J). What is noteworthy is that the expression of *XsHSF10* was up-regulated more than 80 times after 0.5 h treatment with NaHCO3 solution. Moreover, *XsHSF7, XsHSF8,* and *XsHSF11* were up-regulated on NaCl treatment, while their expression displayed first up and then down in 7d under NaHCO$_3$ treatment (Figs. 7E, 7F, 7G).

## Prediction of microRNA target sites in *XsHSFs*

The prediction of microRNA target sites uncovered that the activity of several XsHSF proteins was regulated by microRNAs and that some genes might be regulated by mutiple microRNAs (Data S4). For example, the HSFA group gene *XsHSF2* might be regulated by miR156, miR394, miR395, miR408, miR7129, and miR854, while the miR156, miR171, and miR390 target sites were found in the HSF B group *XsHSF19* mRNA (Table 2). Due to *XsHSF3* and *XsHSF9* have the target sites of miR164 and miR172, respectively. qRT-PCR was performed to determine the expression levels of miR172 and miR164 in the leaves and roots treated with NaCl (Fig. 8). With the increase of NaCl concentration, the expression levels of miR172 firstly increased and then decreased in leaves, and decreased and then increased in roots, which was opposite to the expression level of *XsHSF9* gene, and the correlation coefficient in leaves and root were 0.8791 and 0.6774, respectively. Similarly,

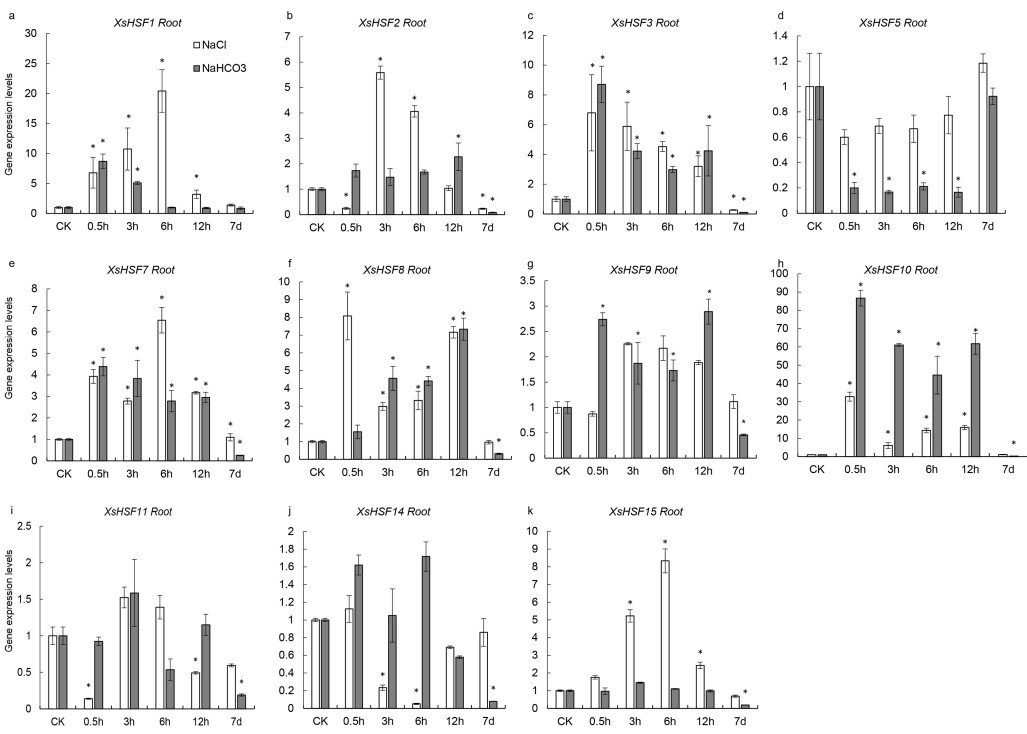

**Figure 7 Expression analysis of fifteen selected *XsHSFs* in roots upon different treatments.** *XsGADPH* was used as an internal control. Vertical lines represent S.E. ($n = 3$), * letters indicate significant differences ($P < 0.05$ by LSD test and fold change $\geq 2$) among treatments. CK, leaf of seedling; *Y*-axis indicate leaf of seedling with 200 mM NaCl or 200 mM NaHCO$_3$ treatment 0.5 h, 3 h, 6 h, 12 h, and 7d, respectively.

the miR172 expression trends in leaves and roots following NaCl concentration were also opposite to those of *XsHSF3*.

## Interaction network of HSF proteins between *X. sorbifolium* and *A. thaliana*

The protein–protein interactions of HSF proteins between *X. sorbifolium* and *A. thaliana* were analyzed using the online software STRING (http://string-db.org) (Fig. 8). The orthologous protein was matched with the highest bitscore, and the lines with different color represented different types of evidence for the interaction. The XsHSF11 and XsHSF13 proteins had highly similar amino sequences to AtHSFA2, and XsHSF9, XsHSF15, XsHSF4, XsHSF12, XsHSF20 had similar sequences to AtHSF3, AtHSFA8, AtHSF1, AtHSFC1, AtHSFA3, respectively (Table 1, Fig. 9). In *Arabidopsis*, *AtHSFA8*, *AtHSFA3*, *AtHSFA2*, At*HSFA4A*, and *AtHSF1C* were reported to regulate heat stress (*Giesguth et al., 2015*; *Friedrich et al., 2021*). AtHSFA3 binds AtHSFA2 and other proteins form heteromeric complexes to acquire thermotolerance. In addition to heat stress, the *HSFA4A* gene was also reported to confer salt tolerance and is regulated by the MAP kinases MPK3 and MPK6 (*Nover et al., 2001*). *AtHSF1a* is the orthologous gene of *XsHSF4* in *Arabidopsis*, and overexpressing the *AtHSF1a* gene can promote the tolerance to diverse stressors, including

**Table 2  miRNA target sites analysis of *XsHSF* genes.**

| Target gene | miRNA |
|---|---|
| *XsHSF1* | miR156, miR167 |
| *XsHSF2* | miR156, miR394, miR395, miR408, miR7129, miR854 |
| *XsHSF3* | miR164, miR477 |
| *XsHSF4* | miR395 |
| *XsHSF7* | miR397, miR858 |
| *XsHSF9* | miR172 |
| *XsHSF15* | miR393, miR530 |
| *XsHSF20* | miR159 |
| *XsHSF21* | miR156 |
| *XsHSF5* | miR156 |
| *XsHSF6* | miR156 |
| *XsHSF8* | miR395 |
| *XsHSF16* | miR156, miR390 |
| *XsHSF17* | miR164 |
| *XsHSF18* | miR854 |
| *XsHSF19* | miR156, miR171, miR390 |

high pH, by promoting inducible HSP expression (*Qian et al., 2014*). Those results showed that *XsHSF* genes might regulate the diverse stresses in *X. sorbifolium*.

# DISCUSSION

*X. sorbifolium* is a special woody oil crop in China. Its fruit oil is a high quality edible oil and the main raw material for biodiesel production. *X. sorbifolium* has a certain tolerance to salt and alkali. However, with the aggravation of soil salinization, it is important to study the salt and alkali tolerance of *X. sorbifolium* to expand the scope of the cultivated region of saline-alkali threat, to relieve the pressure of agricultural production, improve soil quality and restore the ecological environment.

## Characteristics of the HSF family in *X. sorbifolium*

The HSF transcriptional factors play a major role in plant development, biotic, and abiotic stress. The number of *HSF* genes is species-specific, for example, 21, 25, 27, 52, and 56 *HSF* genes were identified in *Arabidopsis,* rice (*Guo et al., 2008*), *poplar* (*Wang et al., 2012*), soybean (*Li et al., 2014*), and wheat (*Ye et al., 2020*), respectively. But the number of HSF family members did not correlate with the genome size of the species, for example, the number of *HSF* genes was similar in *Arabidopsis*, rice, and *X. sorbifolia*, but the genome sizes of *Arabidopsis*, rice, popular and *X. sorbifolia* are vastly different, at 135 Mb (*Goodman, Ecker & Dean, 1995*), 466 Mb (*Yu et al., 2002*), and 420 Mb, respectively. Previous studies showed that the number of plant *HSF* genes may be related to selective pressure during plant evolution in addition to genome replication events (*Klaus-Dieter et al., 2012*). Compared with soybean (52) and wheat (56), our result indicated a decrease of HSF members in *X. sorbifolium*, which may be caused by a lack of whole-genome

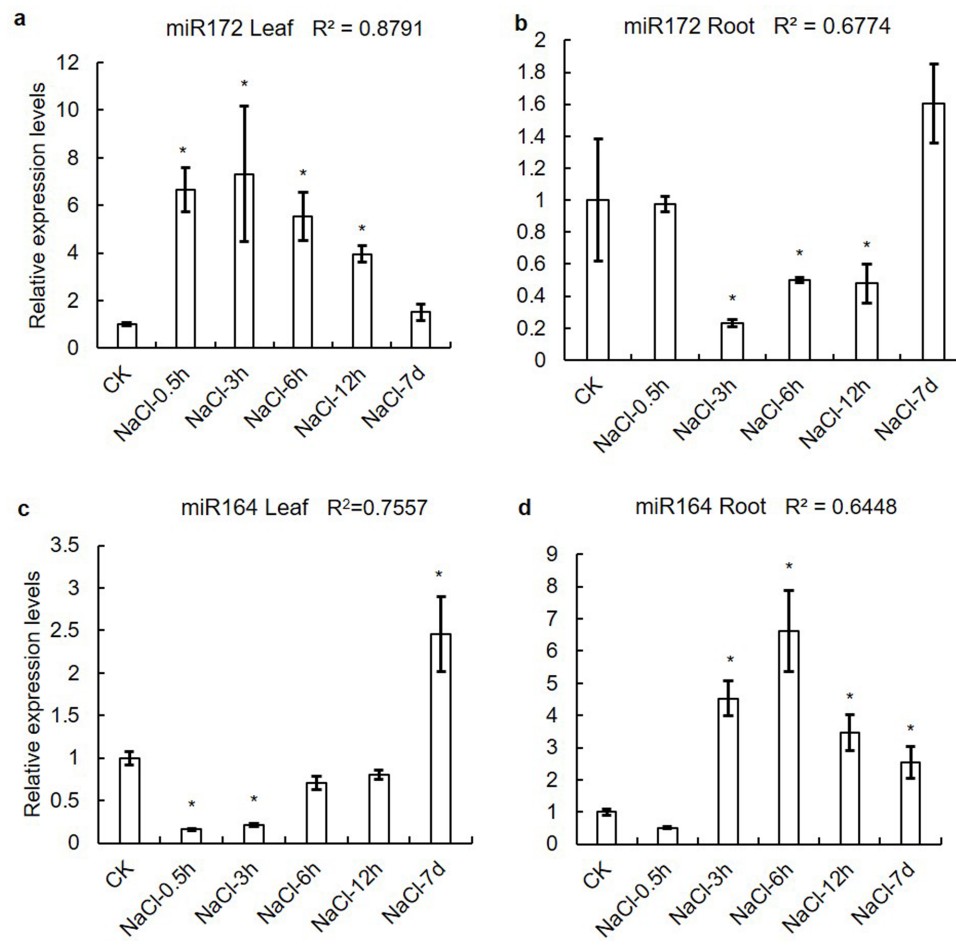

**Figure 8** **Expression levels of miR164 and miR172 in leaves and roots on saline-alkali stress.** *5.8s* was used as an internal control. Vertical lines represent S.E. ($n = 3$), * letters indicate significant differences ($P < 0.05$ by LSD test and fold change $\geq 2$) among treatments. CK, root of seedling; $Y$-axis indicate leaf of seedling with 200 mM NaCl or 200 mM NaHCO₃ treatment 0.5 h, 3 h, 6 h, 12 h, and 7d, respectively.

duplication events in *X. sorbifolium* (*Bi et al., 2019*). In the present study, genes of the *HSF* family were also identified in *X. sorbifolia,* https://www.ncbi.nlm.nih.gov/genome/83651 and https://www.ncbi.nlm.nih.gov/genome/13043, three *Sapindaceae* trees. No recent whole-genome-wide duplication event was detected in *Sapindaceae* (*Lin et al., 2017*), but the number of HSF family members in these species ranged from 15 to 21, indicating that *HSF* genes in *X. sorbifolia,* https://www.ncbi.nlm.nih.gov/genome/83651, and https://www.ncbi.nlm.nih.gov/genome/13043 experienced different selection pressure.

Based on the phylogenetic analysis and the result of the multiple sequences alignment, 21 *XsHSFs* were further clustered into three classes, similar to HSF gene division in other plants (*Busch, Wunderlich & Schöffl, 2005*; *Guo et al., 2008*; *Ma et al., 2014*; *Ma et al., 2015*; *Hu et al., 2015*). The DBD domains of *XsHSF* s were highly conserved, suggesting that the DBD domains were founded before gene functional divergences occurred. Moreover, the number of amino acid residues inserted between HR-A and HR-B in OD domain was

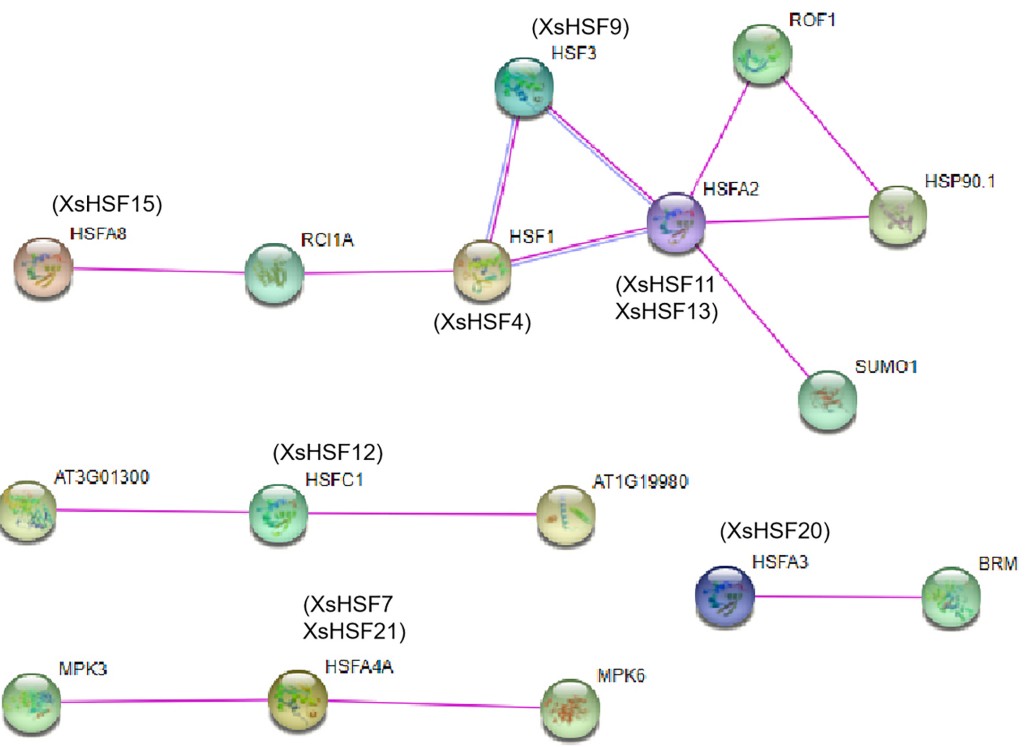

**Figure 9** Interaction networks of seven XsHSF proteins in *X. sorbifolium* according to the orthologs in *Arabidopsis.* The nodes represent different proteins and different colored lines indicate different types of evidence for the interaction. Light blue and rose red lines indicate known interactions originating from curated databases and experimental determinations, respectively.

different among HSFA, B, and C classes, which is in agreement with the results of previous studies (Fig. 4). Several distinct motifs were observed in the three classes, suggested that three class genes underwent different selective constraints and function divergence.

## Response of *XsHSF* genes to saline-alkali stress in *X. sorbifolium*

The HSPs play an important role in response to various biological and abiotic stresses by promoting cell repair, protein folding and degradation, and signal transduction. Moreover, the HSFs are the principal regulatory factors that directly activate the expression of HSPs. The expression of HSFs can be regulated by abiotic stresses, including heat, salt, drought, and cold. But the mechanism of HSF gene response to saline-alkali stress is less studied. Various studies have indicated that different HSF family members have varied functions in regulating plant salinity tolerance. Several HSF genes can enhance the salt tolerance in *Arabidopsis* (*Ogawa, Yamaguchi & Nishiuchi, 2007*), maize (*Jiang et al., 2018*), and *Populus* (*Shen et al., 2015*). AtHsfA6a acts as a transcriptional activator of stress-responsive genes and their overexpression can be induced by exogenous ABA, NaCl, and drought stress (*Hwang et al., 2014*). AtHSFA7b regulates its target genes, containing E-box-like and heat shock element motifs, to trigger serial physiological changes to improve salt tolerance, including adjusting osmotic potential to reduce water loss rate, removing reactive oxygen

species (ROS), and maintaining cellular ion homeostasis (*Zang et al., 2019*). AtHSF4A mediates tolerance to salt and oxidative stress by interacting with mitogen-activated protein kinases MPK3 and MPK6, leading to transcriptional activation of *HSP17.6A* (*Pérez-Salamó et al., 2014*). The expression of maize *ZmHsf04* can increase salt tolerance in *Arabidopsis* (*Jiang et al., 2018*). In addition, ssome HSFs are functioning as a negative regulator, such as ZmHsf08 in maize (*Wang et al., 2021*) and OsHsfB2b in rice (*Xiang et al., 2013*).

The salt tolerant response of *XsHSF* genes in *X. Sorbifolium* is still unclear. In this study, to explore the role of XsHSFs in regulating saline-alkali stress in *X. sorbifolium*, the seedlings were treated with neutral salt NaCl and alkaline salt NaHCO₃. The expression trends of different *XsHSF* gene varied in seedlings treated with a neutral salt and an alkaline salt. The expression profile of 11 genes with low sequence similarity was detected in leaves and roots following salt or alkali treatment. In all, four class A members (*XsHSF10*, *XsHSF11*, *XsHSF15*) and class B members (*XsHSF5*, *XsHSF8*, *XsHSF14*) exhibited high expression levels in leaves on salt treatment, suggesting those genes mainly participated in salt tolerance in leaf (Fig. 6). This is in accord with that the orthologous genes of *XsHSF10* and *XsHSF11*, *AtHSFA2 and AtHSFA1D*, were essential positive regulator for salt tolerance (*Ogawa, Yamaguchi & Nishiuchi, 2007*; *Liu, Liao & Charng, 2011*). On alkaline stress, the expression of most *XsHSFs* were supressed in leaves, including *XsHSF1*, *XsHSF2*, *XsHSF3*, *XsHSF5*, *XsHSF7*, *XsHSF11*, and *XsHSF15*, but only two genes (*XsHSF5* and *XsHSF15*) were also down-regulated in both leaves and roots under alkali stress, suggesting that *XsHSF5* and *XsHSF15* may function as negative regulators in alkali tolerance (Figs. 6 and 7). Most *XsHSFs* expression levels can be induced by alkali stress in less than 12 h, but were suppressed after seven days, indicating that these genes may respond to alkali stress in a short time, but the long-term high concentration of alkali inhibited their expression. Moreover, the functions of XsHSF genes in *X. Sorbifolium* are worthy of further investigation.

## CONCLUSIONS

Analysis of the *X. sorbifolium* genome identified 21 HSF family genes that were characterized, according to their gene structure and protein conserved domains and an expression profile during salinity-alkalinity stress was provided. Based on the phylogenetic tree, these HSF members were classified into three groups and their evolutionary relationships were further analyzed. Furthermore, the expression of 11 *XsHSFs* in different organs (root and leaf) and salinity-alkalinity conditions were changed. Given the likely importance of the HSF family in *X. sorbifolium*, our study provides a valuable basis for the subsequent elucidation of the functions of *XsHSF* genes.

## ACKNOWLEDGEMENTS

The authors would like to thank all the reviewers who participated in the review and MJ Editor for its linguistic assistance during the preparation of this manuscript.

### Funding

The research was supported by the Key R & D plan of Shandong Province (Agricultural Elite Varieties Project) "Breeding of New Breakthrough Varieties of New Woody Oil Tree Species" (2020LZGC00904) Project. The funders had no role in study design, data collection and analysis, decision to publish, or preparation of the manuscript.

### Grant Disclosures

The following grant information was disclosed by the authors:
Key R & D plan of Shandong Province (Agricultural Elite Varieties Project) "Breeding of New Breakthrough Varieties of New Woody Oil Tree Species": 2020LZGC00904.

### Competing Interests

The authors declare there are no competing interests.

### Author Contributions

- Lulu Li conceived and designed the experiments, performed the experiments, analyzed the data, prepared figures and/or tables, and approved the final draft.
- Yiqian Ju performed the experiments, prepared figures and/or tables, and approved the final draft.
- Cuiping Zhang analyzed the data, prepared figures and/or tables, authored or reviewed drafts of the article, and approved the final draft.
- Boqiang Tong analyzed the data, authored or reviewed drafts of the article, and approved the final draft.
- Yizeng Lu analyzed the data, authored or reviewed drafts of the article, and approved the final draft.
- Xiaoman Xie conceived and designed the experiments, analyzed the data, prepared figures and/or tables, and approved the final draft.
- Wei Li conceived and designed the experiments, authored or reviewed drafts of the article, and approved the final draft.

### Data Availability

   The raw measurements are available in the Supplementary Files.

### Supplemental Information

Supplemental information for this article can be found online at http://dx.doi.org/10.7717/peerj.15929#supplemental-information.

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
