# Peer review of "Genome-wide analysis of the heat shock transcription factor family reveals saline-alkali stress responses in Xanthoceras sorbifolium"

_PeerJ, doi:10.7717/peerj.15929_

## Round 0.1 · original submission · Major Revisions

Dear Authors

The manuscript cannot be accepted for publication in its current form. It needs a major revision to be reconsidered for publication. The authors are invited to revise the paper considering all the suggestions made by the reviewers. Please note that requested changes are required for publication.

With Thanks

Reviewer 1 ·

Basic reporting

a. English language. I think that the English language is OK. However, some parts should be reviewed. For instance, ‘were’ should be substituted by ‘was’ (Line 205).

b. References. I think that some aspects of references should be revised: MAJOR CONCERN
i. I miss more updated references. For example, information about protein domains.
ii. Cite format in the text. For example, I believe that this format: (Busch, Wunderlich et al. 2005) is not correct. The correct form will be Busch et al. 2005. Please, confirm with instructions for authors.
iii. I can not find some cites in the reference list.

c. Manuscript structure. I consider that the manuscript should be rearranged. MAJOR CONCERN
i. Introduction is too long. In your title is included ‘saline-alkali stress’ but in your introduction include too details about temperatura, drought, and so on. You could reduce information in the paragraphs: lines 81-123 and 134-167. I think that you should give priority to saline-alkali stress.
ii. The introduction is more extensive than the discussion.
iii. Material and methods. This section may be rearranged: repeated subtitles, methods for other species (P. nume) and gene family (HD-zip),… Please, change this.
iv. I propose a sequence to organize the methodology and results:
1. Identification of gene family.
2. Chromosomal location and duplications.
3. Gene structure and cis-elements.
4. Protein domains and motifs.
5. Phylogenetic analysis.
6. miRNAs.
7. Gene expression.
8. Interaction networks.

d. The manuscript contain all results relevant to the hypothesis.

Experimental design

a. I think that the manuscript is within Aims and Scope of the journal.
b. The research question is well defined and relevant. I think that this research is the first step for demonstrate the functional role of HSF in saline stress, which could be applied to other species.
c. I believe that the research was performed with high technical and ethical standards. However, I think that the mansucript should be extensively rearranged (see comments above).
d. Methods are described with detail but material and methods section should be rearranged (see comments above). MAJOR CONCERN
e. Plant materials. Authors should clarify if leaf buds and young stems were mixed for RNA extraction. This information should be included in the RNA extraction subsection. MAJOR CONCERN

Validity of the findings

I think that it is a good research with a correct justification, all information for replicate included and raw data provided. However, discussion should be improved. For instance, authors should discuss each results considering references.

Additional comments

I give minor comments:
a. Authors use words and numbers for description of genes. For instance, ‘one and three’ or ‘21, 18, 23, and 34’. I think that should use words OR numbers.
b. Web pages should include the date of last access.
c. Line 210. ‘sative’ by ‘sativa’.
d. Line 238. I believe that the correct formo of citation is ‘Tu et al. (2010)’.
e. Tables. Table 1. Arabidopsis homolog or Arabidopsis ortholog? Table 2. Gene names with italics.
f. Some gene names should be written in italics. For example, HSF (line 319). Please, review throught the text. Line 320: poplar without italics.
g. Figures 2-5 shows a black background and the figures can not be visualized very well.
h. Lines 358-362 and 392-399. Where this information is? Authors should be reference tables and figures accordingly.
i. Lines 433-435 and 437-441 are part of the discussion section.
j. Line 479-491. This paragraph lacks of cites.

Reviewer 2 ·

Basic reporting

There are many concerns related to the use of the language; the manuscript must be rewritten using clear English.
different format of references for each part reflects heterogeneous text copying and editing from different sources. Authors should be consistent in the format and the writing style of the journal according to the instructions given to the authors by the journal.
The introduction requires serious revision; references are misplaced and wrongly cited, e.g., Line 101, citation name Yoshida, Nishizawa et al. 2006 have no citation in the references list.
Line 116, citation name (Ogawa, Yamaguchi) in the text, while (Nishiuchi, Daisuke) in the references list.
Line 210, as proposed by Tu et al. (Tu et al. 2010), has no citation in the references list besides the Italic written format.

Experimental design

The material and methods are described in sufficient detail. However, methods should be described with correct and updated information as the referenced genomes are wrong and raise serious doubts and issues with the integrity of the analyzed sequences.

The cited references require serious revision; references are misplaced and wrongly cited, e.g.,
Line 172, Reference genome of Dimocarpus longan was cited as a genome reference by Jiang, Zhang, et al. 2019, while it was Prunus armeniaca L. as the same for Acer yangbiense by Verde, Jenkins et al. 2019.
Line 217, Suyama et al. 2006 has no citation in the references list.
Lines 233 and 235 Zhang et al. 2012 and Zhang et al. 2018 have no citations in the references. list.
Line 245, Chengjie Chen 2018, has no citation in the references.
Line 258, Wang, Pan, et al. 2014 was cited as a full-length XsHSFs sequence according to published microRNA sequences from Citrus sinensis, while in the references list was a different one.

Validity of the findings

Results regarding the phylogenetic tree are wrong, and the grouping is incorrect, with no test of phylogeny nor correct cladistic formations. The phylogenetic tree and all the following data depend on it should be reanalyzed carefully and according to well-known methodology.

Additional comments

I will have to reject the work in its current form. However, authors should be careful while citing references and follow a well-established protocol of gene family analysis published by recent articles from impacted journals.

Reviewer 3 ·

Basic reporting

The basic reporting of the MS is OK.

Experimental design

Experimental design is relevant & meaningful.

Validity of the findings

Can be improved

Additional comments

The manuscript entitled “Genome-wide analysis of the heat shock transcription factor family reveals saline-alkali stress responses in Xanthoceras sorbifolium” by Li et al. investigated the heat shock transcription factor (HSF) family from X. sorbifolium and comprehensively analyzed. The manuscript has some interesting data, but the authors made several mistakes while writing the MS. The manuscript was probably not proofread before submitting.
1. Line 201 ‘The online MEME program was used to display the motif structure of HD-Zip proteins”
2. Line 207 ‘The chromosomal location information of HD-Zip genes was obtained from P. mume genome’.
3. Line 213 ‘DNA sequences of 10-kb upstream and downstream of each HD-Zip gene were searched.
4. Line 213 ‘Transposons and retrotransposons’
5. Line 255 ‘Full-length PmHD-Zips nucleic acid sequences (including introns and UTRs) were obtained’
6. Line 229-236, ‘into the tissue-specific expression patterns of HD-Zip genes; RNA Seq data (GEO No. GSE40162) were used to generate a heat map; HD-Zip gene expression quantities in flower buds; The heat maps were illustrated using the HemI 1.0 software’
7. Line 237-240 ‘238 P. mume cv. ‘Fei Lve’ was cultivated in the greenhouse in JiuFeng National Forest Park of Beijing Forestry University. In order to investigate the expression of PmHBs’’

I could not find related data and results of the above points (1-7) as mentioned in the material method.
8. In line 261, the authors claimed that samples were harvested after 7d of treatment, but the results are shown for 5 d.
9. The revised version should supplement an early response analysis (3, 6, and 12h of the expression) after the treatment.
10. miRNA target site prediction should be verified experimentally for some of the XsHSFs.

---

## Round 0.2 · Major Revisions

Dear Authors

The manuscript still needs a major revision to be reconsidered for publication. The authors are invited to revise the paper considering all the suggestions made by the reviewers. Please note that requested changes are required for publication.

With Thanks

Reviewer 3 ·

Basic reporting

OK

Experimental design

Must be improved.

Validity of the findings

Must be improved.

Additional comments

Overall, the authors have put a significant amount of effort into improving the manuscript. However, I do have concerns regarding the experimental design and verification of results.

Specifically, I find it unclear why the authors chose to use only 0.5h and 12h as time points in their experiment. I suggest that the authors include additional time points, such as 3h and 6h, to provide a more comprehensive understanding of the expression patterns of the genes of interest.

Furthermore, I am concerned that the miRNA target site prediction has not been experimentally verified for some of the XsHSFs. To address this issue, I suggest that the authors verify the expression analysis of a few targets and miRNAs in the same tissue and treatment quantitatively.

In light of these concerns, I recommend that the manuscript be revised to address these issues before it is considered for publication.

---

## Round 0.3 · Minor Revisions

Dear Authors

The manuscript improved after the last round of revision, but the manuscript needs a minor revision to be reconsidered for publication. The authors are invited to revise the paper considering all the suggestions below. Please note that requested changes are required for publication.

Conclusions are not justified. Changes in expression are correlational and do not prove function.

Line 34: " indicating these genes play important roles in salt response. ". Reword to "SUGGESTING that these genes MAY play..."

Line 407 "Furthermore, the expression profiles of 11 XsHSFs in different organs (root and leaf) and salinity alkalinity conditions indicated that they play key roles in salinity-alkalinity tolerance ". It has not been shown that they play key roles. It has been shown that their expression changes. It is not possible to conclude that they play key roles from the data presented here. Delete or reword this.

Line 409, "Given the importance of the HSF family in X. sorbifolium" The importance has not been shown, so this is not a given. Perhaps "Given the LIKELY importance".

With Thanks

---

## Round 0.4 · accepted · Accept

Dear Authors

I am pleased to inform you that after the last round of revision, the manuscript has been improved a lot, and it can be accepted for publication.

Congratulations on the acceptance of your manuscript, and thank you for your
interest in submitting your work to PeerJ.